# Role of Tele-Physical Therapy Training on Glycemic Control, Pulmonary Function, Physical Fitness, and Health-Related Quality of Life in Patients with Type 2 Diabetes Mellitus (T2DM) Following COVID-19 Infection—A Randomized Controlled Trial

**DOI:** 10.3390/healthcare11121791

**Published:** 2023-06-17

**Authors:** Gopal Nambi, Mshari Alghadier, Arul Vellaiyan, Elturabi Elsayed Ebrahim, Osama R. Aldhafian, Shahul Hameed Pakkir Mohamed, Hani Fahad Ateeq Albalawi, Mohamed Faisal Chevidikunnan, Fayaz Khan, Paramasivan Mani, Ayman K. Saleh, Naif N. Alshahrani

**Affiliations:** 1Department of Health and Rehabilitation Sciences, College of Applied Medical Sciences, Prince Sattam Bin Abdulaziz University, Al-Kharj 11942, Saudi Arabia; m.alghadier@psau.edu.sa; 2Department of Nursing, College of Applied Medical Sciences, Prince Sattam Bin Abdulaziz University, Al-Kharj 11942, Saudi Arabia; a.vellaiyyan@psau.edu.sa (A.V.); torabisayed@hotmail.com (E.E.E.); 3Department of Surgery, College of Medicine, Prince Sattam Bin Abdulaziz University, Al-Kharj 11942, Saudi Arabia; o.aldhafian@psau.edu.sa (O.R.A.); aymankamal37@gmail.com (A.K.S.); 4Department of Physical Therapy, Faculty of Applied Medical Sciences, University of Tabuk, Tabuk 71491, Saudi Arabia; s-mohamed@ut.edu.sa (S.H.P.M.); hf_albalawi@ut.edu.sa (H.F.A.A.); 5Department of Physical Therapy, Faculty of Medical Rehabilitation Sciences, King Abdulaziz University, Jeddah 21589, Saudi Arabia; mfaisal@kau.edu.sa (M.F.C.); fayazrkhan@gmail.com (F.K.); 6Department of Rehabilitation Sciences, King Saud Bin Abdulaziz University for Health Sciences, Al Mubaraz 36428, Saudi Arabia; manip@ksu-hs.edu.sa; 7Department of Orthopedic, Faculty of Medicine for Girls, Al-Azhar University, Cairo 11651, Egypt; 8Orthopedic Surgery Department, King Fahad Medical City, Ministry of Health, Riyadh 12231, Saudi Arabia; naifortho@gmail.com

**Keywords:** COVID-19, tele-physical therapy, type 2 diabetes mellitus

## Abstract

The use of tele-rehabilitation (TR) in type 2 diabetes mellitus (T2DM) following COVID-19 infection remains unexplored. Hence, the purpose of this study was to determine the clinical effects of tele-physical therapy (TPT) on T2DM following COVID-19 infection. The eligible participants were randomized into two groups, a tele-physical therapy group (TPG; *n* = 68) and a control group (CG; *n* = 68). The TPG received tele-physical therapy for four sessions a week for eight weeks, and the CG received patient education for 10 min. The outcome measures were HbA1c levels, pulmonary function (forced expiratory volume in the first second (FEV1), forced vital capacity (FVC), FEV1/FVC, maximum voluntary ventilation (MVV), and peak exploratory flow (PEF)), physical fitness, and quality of life (QOL). The difference between the groups in HbA1c levels at 8 weeks was 0.26 (CI 95% 0.02 to 0.49), which shows greater improvement in the tele-physical therapy group than the control group. Similar changes were noted between the two groups after 6 months and at 12 months resulting in 1.02 (CI 95% 0.86 to 1.17). The same effects were found in pulmonary function (FEV1, FVC, FEV1/FVC, MVV, and PEF), physical fitness, and QOL (*p* = 0.001). The reports of this study show that tele-physical therapy programs may result in improved glycemic control and improve the pulmonary function, physical fitness, and quality of life of T2DM patients following COVID-19 infection.

## 1. Introduction

Diabetes mellitus (DM) is a serious health condition in which the body is unable to metabolize carbohydrates either due to lack of insulin production or abnormal insulin signaling. In recent times, diabetes is considered a global healthcare concern because 537 million adults (20–79 years) are living with diabetes; that is to say that 1 in every 10 people suffer from it. This number is predicted to reach 643 million by 2030 and 783 million by 2045 [1]. Generally, there are two variants of diabetes: type 1 DM (T1DM) and type 2 DM (T2DM), of which T2DM is the more prevalent type in the adult population [2]. The healthcare expenditure of patients with T2DM are twofold more than that of people without diabetes mellitus [3]. Moreover, T2DM is a leading cause of cardiovascular and metabolic diseases [4]. It is the second-most common non-infectious condition in individuals presenting with coronavirus disease 2019 (COVID-19) infection (prevalence = 10.0%), and is highly correlated with disease severity [5]. COVID-19 infection causes T2DM patients to exhibit the following symptoms, impacting their general health: altered pulmonary functions, lower exercise tolerance, decreased muscular strength, cognitive impairments, and aberrant psychosomatic behavior [6]. In addition, these changes negatively influence their physical efficiency and health-related quality of life (HRQOL). These changes require appropriate medical care and an effective physical therapy training program that can reduce the long-term consequences of T2DM following COVID-19 infection.

To prevent or delay the onset and severity of these clinical features and the associated consequences of T2DM, regular exercise and physical training is suggested either alone or in combination with diet modification [7]. Due to the risk of spreading COVID-19 infection, regular physical training in an outpatient clinic during the pandemic has become difficult due to social distancing. Compliance and regularity in following the physical therapy training program is the main drawback of the exercise program [8].

The advancements in science and technology in the field of computer science found innovations in the healthcare sector [9]. In recent times, tele-rehabilitation (TR) has used digital communication systems to deliver healthcare to patients [10]. Tele-rehabilitation is a valuable method to monitor and provide treatment for remotely located patients and it can maintain treatment compliance during the COVID-19 pandemic [11]. Controlling blood glucose and blood pressure (BP) levels, managing weight and prescribing diet patterns can be accomplished through tele-rehabilitation-based approaches [12]. Patient education on diabetic care is given through online webinars, virtual meetings, and personal phone calls and they are supervised and encouraged to perform exercises and physical activities. The patient and their treating therapists can discuss the type, mode, and frequency of the exercise program and its follow-up regime, which enhances compliance with the treatment and maintains the follow-up regimen. It trains the patient to take responsibility for following their rehabilitation program [13].

TR systems make use of mobile phones, computers, virtual conferences, and satellite technologies for follow-ups to prevent further consequences of diabetes mellitus [14]. Recent research investigating tele-rehabilitation and telehealth on T2DM patients investigated the blood glucose levels and other associated features and they show that the TR approach improved the blood glucose and other biochemical levels of their patients significantly [15,16]. Furthermore, physical therapy training programs using tele-rehabilitation for T2DM patients following COVID-19 remains unexplored and investigated, though tele-rehabilitation has already been applied to orthopedic problems, neurological diseases and cardio-respiratory disorders [17,18]. There is an urgent need for tele-physical therapy (TPT) training during this COVID-19 outbreak to manage and prevent the consequences of diabetes. Hence, the objective of the present study was to determine the effects of tele-physical therapy on glycemic control, pulmonary function, physical fitness, and HRQOL in patients with T2DM following COVID-19 infection. We hypothesized that 8 weeks of supervised tele-physical therapy with 12 months of follow-up can positively influence patients with T2DM following COVID-19 infection. The results of this study will help health care professionals including physiotherapists, respiratory therapists, and general physicians in adapting clinical practice models to utilize Tele-Health therapy for patients living with T2DM and symptoms of COVID-19 infection.

## 2. Methods

### 2.1. Study Design

A Prospective, two-arm blinded, Randomized Control Trial was conducted between January 2021 and September 2022. The participants were screened by a physician at the hospital of Prince Sattam bin Abdulaziz University, Saudi Arabia as per the diabetes mellitus diagnostic criteria defined by the ICD (International Classification of Disorders). The trial was conducted at this university and ethical approval was obtained from the Department Ethical Committee (DEC) with the reference number RHPT/021/085. The study protocol and its informed consent forms were approved by the DEC. The study strictly followed the ethical guidelines laid down by the 1964 Declaration of Helsinki and was registered retrospectively in the clinical trial registry with NCT05599893 on 30 October 2022.

### 2.2. Participants

The participant list was extracted from the hospital records of King Khalid Hospital and University Hospital, Al Kharj, Saudi Arabia. They were contacted by telephone first and if they expressed an interest to participate in the study, then they were given an appointment for the initial visit. Participants between 18 and 60 years of age who were diagnosed with one-month post COVID-19 infection with mild dyspnea and T2DM (taking Metformin 500 mg orally twice a day) and who could use smart phones were selected to be a part of the study. Participants who had post COVID-19 symptoms, hospitalized for COVID-19 infection, neurological (radiculopathy, myelopathy, and disc problems) or orthopedic problems, cardiopulmonary diseases (stroke, hypertension, and syncope), other metabolic or endocrinal problems, metastasis, pregnancy, any contraindications to physical therapy exercises (fracture, instability, osteoporosis, arthropathy, and neural symptoms), or cognitive and mental disorders, as well as those taking analgesics or corticosteroids and doing regular physical training or involving in any exercise training, were excluded. The flow of the study program was recorded per the consolidated standards of reporting trials (CONSORT) guidelines and is shown in Figure 1.

### 2.3. Randomization

Prior to participants’ enrollment in the study, a general physician evaluated 247 participants according to the eligibility criteria to identify those suitable to be included in the trial. Of the 247 participants, 136 were randomized into either of two groups, the tele-physical therapy group (TPG; *n* = 68) and the control group (CG; *n* = 68), through computer generated random number method and the subject numbering was given consecutively as per their enrollment in the study.

### 2.4. Blinding

To ensure allocation concealment, an independent therapist randomly picked up an envelope that contained patient information and in a blinded fashion, segregated the patient into either of these two groups. The treating or supervising therapists could not be blinded due to practical feasibility. Assessing therapists who are measuring the outcomes at different intervals of each participant were blinded. Participants were unaware of the assigned treatment and were also asked to not disclose their assignment to the assessing therapists and co-participants at any time during the study.

### 2.5. Intervention

All study participants had to sign a written informed consent form before the start of the study. Three physical therapists with up to fifteen years of clinical experience of treating patients living with T2DM were assigned to the tele-physical therapy group. To reduce the variability between the therapists (intervention bias), they were instructed to follow the strict guidelines of the study protocol approved by the ethical committee. Participants in the TPG underwent tele-physical therapy sessions, which included an internet-based video conference under the supervision of physical therapists. The participants attended the presentation from the comfort of their homes using a mobile application called RehabApp, except for the first session, in which the training was carried out at the clinic for the participants to learn and understand the exercises. This application alerted the participants to perform the exercise and monitored the exercises throughout the session. It also recorded their heart rate, the total duration of exercise, and any other interventions taken during the training.

The training session included 10 min of breathing exercises, 30 min of aerobics, and 40 min of resistance exercises that included 15 different exercises involving the gradual strengthening and stretching of the lower and upper extremity muscles, which is described in Table 1.

During the 1st and 2nd weeks, the 3rd and 4th weeks, the 5th and 6th weeks and the 7th and 8th weeks the repetition of exercises was increased from 10–15 to 15–20, 20–25, and 25–30 times per session, respectively. Each aerobic training session consisted of 5 min of warm-up, 20 min of aerobic training and 5 min of cool down. In the same way, each resistance exercise session consisted of 5 min of warm-up, 30 min of training, and 5 min of cool down. The participants in the TPG received training four times a week for 8 weeks and each session lasted for 80 min. The participants themselves monitored their heart rate, blood pressure and SpO_2_ before and after each session using a digital sphygmomanometer and pulse oximeter. Participants were provided education and instruction on device use, this was delivered by the physiotherapist at their initial session to not only ensure patient safety, but also to improve self-monitoring adherence and repeated measure reliability. The intensity of exercise for each participant was adjusted based on their Modified Borg Scale for dyspnea or fatigue. The participant was advised to take a rest if their perceived exertion rate was >7 on the Borg Scale [19]. After 8 weeks of training, patients were observed continuously at 6 and 12 months. During the first visit, participants in the control group (CG) received patient education about the causes, risk factors, complications, and required lifestyle modifications for 10 min from physical therapists and also received a printed pamphlet with detailed instructions to be followed at home. They were informed to follow their routine, refrain from being sedentary, perform routine household activities, maintain a balanced diet, and get adequate sleep (6–8 h per day).

### 2.6. Outcome Measures

The sociodemographic characteristics, durations of disease, and HbA1c levels of all the study participants were measured. To know the immediate, short- and long-term effects the outcome variables were measured at the beginning of the study, after 8 weeks, 6 months, and at 12 months follow up.

#### 2.6.1. Primary Outcome

Glycemic level: The hemoglobin A1C test, also known as the HbA1c test, is the fraction of glycated hemoglobin (Hb) in the erythrocytes, which correlates with blood glucose levels 3 months prior and it is a reliable biomarker method for the diagnosis of T2DM and also ongoing diabetes control [20]. The blood sample was collected from the tip of the finger through finger-pricking, and the sample was analyzed with a kit (HbA1c FIA test, CTK Biotech, Poway, CA, USA). A normal HbA1c level is less than 5.6%; scores between 5.7% and 6.4% are considered prediabetic, and scores of 6.5% or higher are considered diabetic. 

#### 2.6.2. Secondary Outcome

Pulmonary function: A portable spirometer device (Spiro lab, MIR, Rome, Italy) was used to measure pulmonary function. The participant was asked to take a deep breath and blow into a tube connected to the spirometer device. The spirometer measured the forced expiratory volume in the first second (FEV1), forced vital capacity (FVC), FEV1/FVC, maximum voluntary ventilation (MVV), and peak exploratory flow (PEF). It is a reliable, valid, and reproducible tool used to measure pulmonary function [21].

Physical fitness: Fitness was measured with a six-minute walk test (6MWT), which measures the functional exercise capacity of the participants. A 30 m-long flat corridor was used for the test. The subject was asked to walk down the corridor at their normal pace for 6 min and the test was repeated after 10 min. The longest 6MWT distance (in meters) covered was used for analysis. This test is simple to administer and a reliable tool for assessing cardiorespiratory fitness in subjects with T2DM [22].

Health-related quality of life (HRQOL): HRQOL was measured with the Short Form Health Survey-12 (SF-12). Physical component score (PCS) and mental component score (MCS) were reported, with a higher score indicating better health. The SF-12 was the most commonly used validated questionnaire among patients with T2DM. Furthermore, the SF-12 was the most reliable, easiest to use, least time-consuming, and most valid tool to measure HRQOL in T2DM patients [23].

### 2.7. Sample Size

Sample size was calculated for an experimental study design (randomized control trial) and the outcome measure used was HbA1c. The alpha error score was set to (α = 0.05) and the power was set to (1 − β (0.15) = 0.85). To find a minimal clinically important change (MCID) of 1.06 units and a standard deviation (SD) of 0.31 units in a two tailed parametric z-test, the required participants in each group were 58. The expected dropout rate was set to 15% and the power calculation showed 68 participants in each group [24].

### 2.8. Statistical Analysis

The reports are presented as means and standard deviations (SD) with a 95% confidence interval (CI) and with an upper and lower limit. The Kolmogorov–Smirnov test was performed to find the normality of the data and to decide the tests to be used for analysis. The study and the statistical tests were performed and presumed to follow the intention to treat analysis procedure. Therefore, all the participants who are randomized are included in the statistical analysis and analyzed according to the group they were originally assigned, regardless of what treatment they received, degree of compliance to the treatment, and drop out. The time and group (4 × 2) mixed model for repeated measures (MMRM) of primary and secondary outcome measures was used among TPG and CG groups at various intervals. The parametric z-test was used to find between group effects, and the repeated measure analysis of variance (rANOVA) was used to find intra-group effects. The α level was set at 0.05, and IBMSPSS—online version 20 was used.

## 3. Results

In total, 247 participants were assessed for eligibility with 136 deemed eligible and randomized to TPG (*n* = 68) and CG (*n* = 68) those excluded in the trial can be seen in Figure 1. Thirty-six subjects had experience with previous physical therapy interventions or currently under physical therapy training, twenty-nine had other systemic problems, six subjects had musculoskeletal injuries, eight participants underwent surgery in their recent past, and thirty-two participants did not agree to participate in the trial; all of these subjects were excluded from the study. Finally, the one hundred and thirty-six remaining participants were selected and randomly separated into the study groups. Four subjects in TPG and five subjects in CG did not finish the further follow-up measurements due to personal reasons and time constraints. Table 2 provides the demographic characteristics of the study participants and the Kolmogorov–Smirnov test was performed, which demonstrates groups were matched for gender, height, weight, BMI, a1c, BP, and disease duration. The data of primary and secondary outcome measures were presented as means and SDs with a 95% confidence interval (CI) with an upper and lower limit.

Table 3 reports the pre and post treatment mean and standard deviation of each group over time. The mixed model repeated measures (4 × 2) MMRM of the primary outcome measure (glycemic level (HbA1c)) show a statistically significant difference (*p* < 0.001) among TPG and CG groups at baseline, 8 weeks, 6 months, and at 12 months. The baseline measurement of HbA1c did not show any significant difference (*p* > 0.05). At the same time, measurement at 8 weeks reported advancement of 0.26 percentage points higher HbA1c (95% CI 0.02 to 0.49) in the TPG than in the CG. The same changes were observed after 6 months of follow-up. Again, at 12 months’ measurement, there was a significant difference of 1.02 percentage points higher HbA1C (95% CI 0.86 to 1.17) (*p* < 0.001) in the TPG than in the CG, which is shown in Table 3 and Table 4.

The mixed model repeated measures (4 × 2) (MMRM) of the secondary outcome measures (FEV1, FVC, FEV1/FVC, MVV, PEF, 6MWT, and QOL) were calculated to report statistically significant difference (*p* < 0.001) among TPG and CG groups at baseline, 8 weeks, 6 months, and at 12 months. The baseline measures did not show any statistical differences (*p* > 0.05). At the same time, measurements at 8 weeks (FEV1 −0.1 (95% CI −0.13 to −0.06), FVC −0.17 (95% CI −0.22 to −0.11), FEV1/FVC −2.3 (95% CI −3.91 to −0.68), MVV −8.9 (95% CI −10.56 to −7.23), PEF −0.6 (95% CI −0.70 to −0.49), 6MWT −20.4 (95% CI −33.0 to −7.8), and QOL −6.2 (95% CI −7.79 to −4.60)) show statistically significant differences (*p* < 0.001) among TPG and CG groups. Similar changes were noted after 6 months of intervention and again after 12 months (FEV1 −1.7 (95% CI −3.84 to 0.44), FVC −1.43 (95% CI −1.50 to −1.34), FEV1/FVC −12.1 (95% CI −14.15 to −10.04), MVV −13.7 (95% CI −15.53 to −11.86), PEF −0.9 (95% CI −1.02 to −0.77), 6MWT −37.0 (95% CI −50.7 to −23.4) and QOL −18.9 (95% CI −21.17 to −16.62)) that show the significant differences (*p* = 0.001) between the TPG and the CG. The statistical reports show improvement for the secondary variables in the TPG vs CG at 12 months’ follow-up, which is displayed in Table 3 and Table 4. The Cohen’s effect size of FEV1 (d = 8.78), FVC (d = 6.50), FEV1/FVC (d = 2.11), MVV (d = 2.34), PEF (d = 2.50), 6MWT (d = 0.92), and QOL (d = 2.84) depict greater effects in the TPG than in the CG. The graphical representation of all of the outcome variables between the two groups is shown in Figure 2.

## 4. Discussion

This randomized controlled study found that tele-physical therapy on T2DM patients following COVID-19 infection effectively improved the patients’ glycemic control, pulmonary function status, physical fitness, and HRQOL following a 12-month follow-up. Based on patient responses, it was observed that holding TPT sessions four times a week was most effective in motivating the patients in the experimental group (TPG). A high rate of treatment compliance (96%) was seen in the study; patients in the tele-physical therapy group, on average, participated in 31 out of the 32 sessions and the most common reason for non-attendance was personal reasons (out of station, sick, and tired). We noted that patients with T2DM performed breathing exercises, and aerobic and resistance exercises, which included upper and lower extremity exercises and trunk exercises with a gradual increase in intensity based on the participants’ performance. These exercises improve metabolic features and insulin sensitivity and reduces abdominal fat in T2DM patients. While performing the resistance exercises, no extra load was added and the patient’s body weight; speed of exercises and number of repetitions were the factors that influenced exercise difficulty. In addition, no special devices were required to perform these exercises.

Types of physical rehabilitation therapies delivered through tele-rehabilitation include strengthening exercises, motor retraining, goal setting, virtual reality, robotic therapy, and community-therapy. The tele-physical therapy-based motor strengthening exercises are the most commonly used modality in the field of rehabilitation and to our knowledge, we are the first to study the effects of TPT on patients with T2DM following COVID-19 infection. Previous systemic reviews and meta-analyses reported that supervised physical exercises reduced HbA1c levels and enhanced physical fitness in patients with T2DM [25,26]. The Cohen’s effect size of HbA1c (d = 2.21) depicts a greater effect in the tele-physical therapy group than in the control intervention group. This is in agreement with our study’s reports after the tele-physical therapy program. Thus, our observed reduction in HbA1c levels might be expected to produce a reduction in cardiovascular disease risk and microvascular complications in patients with T2DM. Maillard F and colleagues stated that although this effect is limited, it emphasizes the requirement of supervised physical exercise interventions to improve physical fitness [27].

There is limited evidence on the effects of tele-physical therapy on pulmonary function in T2DM patients following COVID-19 infection. A recent meta-analysis revealed that diaphragmatic breathing exercises were effective in maintaining and/or improving pulmonary functions such as FEV1, FVC, FEV1/FVC, MVV, and PEF in chronic obstructive pulmonary disease (COPD) patients [28]. Similar effects were noted in our experimental group. Based on our knowledge, there is no literature available on the effects of tele-physical therapy on the pulmonary function and physical fitness of diabetic patients following COVID-19 infection; therefore, this is the first study to show that a tele-physical therapy program can enhance the pulmonary function, physical fitness and QOL of patients with T2DM following COVID-19 infection.

Low physical fitness level is a major risk factor for all-cause mortality in T2DM; hence, fitness exercises are an integral part of diabetic care [29]. Studies on the elderly population show improved physical fitness levels when telephone-based exercise counselling is used [30]. In our training program, we observed a 37m average improvement in 6MWT in the tele-physical therapy group when compared to the control group at 12-month follow-up. However, in the literature, there are inconsistent results on the effectiveness of tele-rehabilitation on improving 6MWT for patients with chronic diseases. A study showed no significant difference in the 6MWT between the TR group and the control group in chronic respiratory disease patients [31]. The reasons for the contradictory results in 6MWT would be inconsistency in the tele-monitoring and exercise training procedures. Piotrowicz E. et al. observed that a significant improvement in 6MWT was seen in a center-based treatment group than in a TR group in patients with heart failure [32].

In a review, Klonoff D.C. stated that tele-rehabilitation was a promising tool in the field of diabetic care by improving communication between the physical therapist and patient, thereby improving the patient’s quality of life [33]. Hence, we believed that a similar result could be achieved when TPT was applied to patients with diabetes. Our study’s findings show that TPT significantly improved patients’ health-related quality of life. Even though tele-physical therapy is proven to be effective as well as cost-effective, patients still hesitate to participate in such programs due to the reduced interaction between the therapist and patients [12]. It was also seen that those living with depression were less likely to participant in such programs, which clearly identifies depression is an indicator of low engagement with clinical therapies of any type [34].

One of the greatest issues in treating patients with T2DM is the high cost of medical treatment [35]. TPT is one rehabilitation program that can effectively bring down the cost of treatment. It is a reasonable and effective additional and/or alternative form of treatment compared to traditional therapies. The use of well-regulated exercise programs, where all exercises are conducted at home with careful monitoring of training, is another strength of this study. Our findings are generalizable to T2DM patients following COVID-19 infection, because the group was diverse in terms of age, gender, drug use, and comorbidities. We achieved good treatment compliance and a low dropout rate despite a group with numerous medical issues. A few limitations were noted while executing the study. First, especially in the older age groups, some hurdles that restrict the use of tele-physical therapy are lack of access to the internet and fear of using smartphones. Second, patients with past histories of COVID-19 infections with mild dyspnea symptoms were included; hence, the results cannot be generalized to patients with no, moderate, or severe dyspnea. Third, the treatment provided to the tele-physical therapy and control intervention groups were not similar, which could have affected the results. Fourth, our study included only a tele-physical therapy group and a control intervention group; therefore, including another group having physical therapy training at the physical therapy department would be helpful in determining the effects of tele-physical therapy on T2DM patients following COVID-19 infection. Furthermore, the difference in tele-rehabilitation effects in type 2 diabetic patients taking metformin in comparison to those who did not taking metformin could be recommended for future study to find the additional effects.

## 5. Conclusions

This study’s findings show that tele-physical therapy programs can result in improved glycemic control, pulmonary function, physical fitness, and QOL of T2DM patients following COVID-19 infection. Hence, incorporating a tele-physical therapy program in combination with current pharmacological and/or non-pharmacological treatment would be beneficial in managing patients with T2DM following COVID-19 infection, especially those who do not have access to clinic-based exercise programs. In addition, diaphragmatic breathing exercises are an effective substitute for conventional therapy for correcting pulmonary health in T2DM patients following COVID-19 infection. As these exercises are easy to perform and require minimal equipment, they are appropriate for home-based exercise programs under the supervision of a trained physical therapist.

## Figures and Tables

**Figure 1 healthcare-11-01791-f001:**
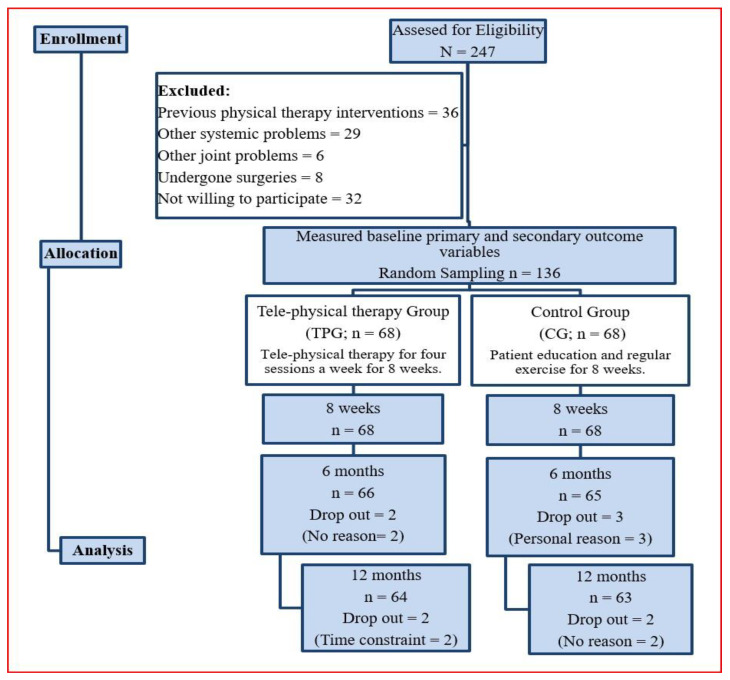
Flow chart showing the study details.

**Figure 2 healthcare-11-01791-f002:**
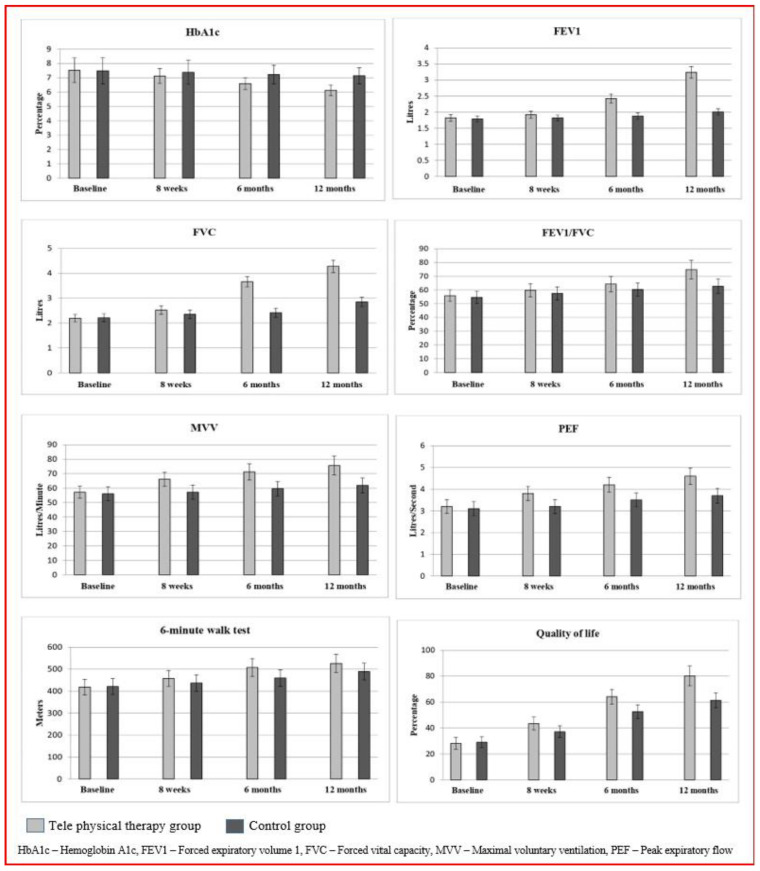
Mean and standard deviation scores of primary and secondary outcomes of tele-physical therapy and of control intervention group.

**Table 1 healthcare-11-01791-t001:** Table showing the exercise interventions of tele-physical therapy group.

Exercise Type	Description	Duration	Intensity	Frequency
Breathing exercise	Diaphragmatic breathing exercise: Step 1: The patient sits down on a chair and relaxes the upper chest and shoulders. Both hands are positioned on the abdomen. The patient should breathe slowly and deeply. Upon inhaling and exhaling, the hands should be felt to move out and in. Step 2: The patient breathes slowly and deeply from the diaphragm and upon inhaling maintains pressure with the hands to provide resistance on the abdomen.	Total: 10 min 4 min plus 1 min break between sets.	12 reps/set for 2 sets.	4 sessions per week for 8 weeks.
Aerobic exercise	Moderate intensity aerobic exercises: Brisk walking or running outside near home or on a treadmill at home with warm up comprising walking at regular speed and cool down comprising seated muscle stretching exercises.	Total: 30 min 20 min plus 5 min warm up and 5 min cool down.	Target heart rate (THR): 40–60%	4 sessions per week for 8 weeks.
Resistance exercise	Supine position: Reciprocal straight leg raise Reciprocal hip flexion and extension Trunk flexion Side lying position: 4. Hip abduction Sitting with legs outstretched position: 5. Hamstring muscle stretch Chair exercises: 6. Shoulder/Chest stretching (with hands on waist) 7. Shoulder/Chest stretching (with hands clenched behind back) 8. Shoulder elevation 9. Shoulder circles Standing exercises: 10. Shoulder flexion/extension 11. Shoulder abduction/adduction 12. Reciprocal lateral trunk flexion/extension 13. Reciprocal hip and knee flexion/extension 14. Quarter knee bends 15. Reciprocal reach upwards with hands	Total: 40 min 2 min each exercise. 30 min plus 5 min warm up and 5 min cool down.	1–2 weeks: 10–15 repetitions. 3–4 weeks: 15–20 repetitions. 5–6 weeks: 20–25 repetitions. 7–8 weeks: 25–30 repetitions. 2 sets with 30 s break.	4 sessions per week for 8 weeks.

**Table 2 healthcare-11-01791-t002:** The demographic characteristics of the TPG and CG groups.

Sr. No	Variable		TPG (*n* = 68)	CG (*n* = 68)	*p*-Value
1	Age (year)	-	48.6 ± 4.2	47.8 ± 4.5	0.285 *
2	Gender	Male	33 (49%)	32 (47%)	-
		Female	35 (51%)	36 (53%)	-
3	Height (cm)	-	165.2 ± 4.2	166.3 ± 3.9	0.115 *
4	Weight (kg)	-	77.89 ± 4.3	78.01 ± 4.2	0.869 *
5	BMI (kg/m^2^)	-	24.2 ± 2.13	24.4 ± 2.32	0.601 *
6	HbA1c (%)	-	7.52 ± 0.86	7.48 ± 0.91	0.792 *
7	Blood pressure (mm/Hg)	Systolic	128.2 ± 7.21	127.2 ± 7.55	0.431 *
8		Diastolic	91.2 ± 5.32	92.0 ± 5.64	0.396 *
9	Disease duration (years)	-	7.84 ± 0.92	7.92 ± 0.89	0.607 *
10	Disease severity	Severe	8 (12%)	10 (15%)	-
		Non-severe	60 (88%)	58 (85%)	-
11	Co-morbidity	Heart disease	6 (9%)	5 (7%)	-
		Hypertension	23 (34%)	18 (26%)	-
		Obesity	12 (18%)	9 (13%)	-
		Lung disease	5 (7%)	3 (4%)	-
		Other problems	9 (13%)	6 (9%)	-
12	Smoking	Yes	22 (32%)	24 (35%)	-
		No	46 (68%)	44 (65%)	-
13	Medications	Yes	48 (71%)	51 (75%)	-
		No	20 (29%)	17 (25%)	-
14	Insulin	Yes	18 (26%)	15 (22%)	-
		No	50 (74%)	53 (78%)	-

* Non-significant; TPG—tele-physical therapy group; CG—control group.

**Table 3 healthcare-11-01791-t003:** Pre- and post-treatment mean and standard deviation measures of the TPG and CG group.

Sr. No	Variable		TPG (*n* = 68)	CG (*n* = 68)	*p*-Value
1	HbA1c (Percentage)	Base line	7.52 ± 0.86	7.48 ± 0.91	0.792 *
		8 weeks	7.12 ± 0.52	7.38 ± 0.84	0.031
		6 months	6.59 ± 0.41	7.22 ± 0.65	0.001
		12 months	6.12 ± 0.36	7.14 ± 0.56	0.001
		*p*-value	0.001	0.001	
2	FEV1 (liters)	Base line	1.82 ± 0.10	1.79 ± 0.09	0.068 *
		8 weeks	1.92 ± 0.11	1.82 ± 0.09	0.001
		6 months	2.42 ± 0.14	1.88 ± 0.10	0.001
		12 months	3.24 ± 0.18	2.01 ± 0.10	0.001
		*p*-value	0.001	0.001	
3	FVC (liters)	Base line	2.19 ± 0.15	2.21 ± 0.16	0.453 *
		8 weeks	2.52 ± 0.17	2.35 ± 0.17	0.001
		6 months	3.66 ± 0.21	2.41 ± 0.18	0.001
		12 months	4.28 ± 0.25	2.85 ± 0.19	0.001
		*p*-value	0.001	0.001	
4	FEV1/FVC (Percentage)	Base line	55.9 ± 4.2	54.6 ± 4.4	0.080 *
		8 weeks	59.8 ± 4.8	57.5 ± 4.7	0.005
		6 months	64.3 ± 5.6	60.3 ± 4.9	0.001
		12 months	74.8 ± 6.8	62.7 ± 5.2	0.001
		*p*-value	0.001	0.001	
5	MVV (liters/minute)	Base line	57.2 ± 4.21	56.1 ± 4.81	0.158 *
		8 weeks	66.1 ± 4.89	57.2 ± 4.92	0.001
		6 months	71.2 ± 5.63	59.6 ± 5.05	0.001
		12 months	75.6 ± 6.45	61.9 ± 5.21	0.001
		*p*-value	0.001	0.001	
6	PEF (liters/second)	Base line	3.2 ± 0.31	3.1 ± 0.32	0.066 *
		8 weeks	3.8 ± 0.32	3.2 ± 0.32	0.001
		6 months	4.2 ± 0.34	3.5 ± 0.32	0.001
		12 months	4.6 ± 0.38	3.7 ± 0.34	0.001
		*p*-value	0.001	0.001	
7	6-min walk test (meters)	Base line	418.18 ± 35.4	421.12 ± 36.3	0.633 *
		8 weeks	457.44 ± 36.7	436.98 ± 37.4	0.015
		6 months	507.31 ± 40.2	459.79 ± 37.9	0.000
		12 months	526.28 ± 42.3	489.22 ± 38.1	0.000
		*p*-value	0.001	0.001	
8	Quality of life (SF-12)	Base line	28.2 ± 4.5	29.1 ± 4.1	0.224 *
		8 weeks	43.4 ± 4.9	37.2 ± 4.5	0.001
		6 months	64.1 ± 5.7	52.5 ± 5.4	0.001
		12 months	80.2 ± 7.6	61.3 ± 5.7	0.001
		*p*-value	0.001	0.001	

* Non-significant; TPG—Tele-physical therapy group; CG—control group; HbA1c—hemoglobin A1c; FEV1—forced expiratory volume 1; FVC—forced vital capacity; MVV—maximal voluntary ventilation; PEF—peak expiratory flow; SF-12—short form 12.

**Table 4 healthcare-11-01791-t004:** Pre- and post-treatment mean difference and confidence interval (upper limit and lower limit) scores of TPG and CG group.

Variable/Time		Baseline	8 Weeks	6 Months	12 Months
		Mean Difference CI95% (Upper Limit—Lower Limit)			
HbA1c	TPG × CG	−0.04 (−0.34 to 0.26)	0.26 (0.02 to 0.49)	0.63 (0.43 to 0.82)	1.02 (0.86 to 1.17)
	*p*-value	0.7928 *	0.031	0.001	0.001
FEV1	TPG × CG	−0.03 (−0.06 to 0.00)	−0.1 (−0.13 to −0.06)	−0.5 (−0.58 to −0.49)	−1.7 (−3.84 to 0.44)
	*p-*value	0.068 *	0.001	0.001	0.119 *
FVC	TPG × CG	0.02 (−0.03 to 0.07)	−0.17 (−0.22 to −0.11)	−1.25 (−1.31 to −1.18)	−1.43 (−1.50 to −1.34)
	*p-*value	0.453 *	0.001	0.001	0.001
FEV1/FVC	TPG × CG	−1.3 (−2.75 to 0.15)	−2.3 (−3.91 to −0.68)	−4.0 (−5.78 to −2.21)	−12.1 (−14.15 to −10.04)
	*p-*value	0.080 *	0.005	0.001	0.001
MVV	TPG × CG	−1.1 (−2.63 to 0.43)	−8.9 (−10.56 to −7.23)	−11.6 (−13.4 to −9.78)	−13.7 (−15.53 to −11.86)
	*p-*value	0.158 *	0.001	0.001	0.001
PEF	TPG × CG	−0.1 (−0.20 to 0.00)	−0.6 (−0.70 to −0.49)	−0.7 (−0.81 to −0.58)	−0.9 (−1.02 to −0.77)
	*p-*value	0.066 *	0.001	0.001	0.001
6-min walk test	TPG × CG	2.94 (−9.2 to 15.1)	−20.4 (−33.0 to −7.8)	−47.5 (−60.7 to −34.2)	−37.0 (−50.7 to −23.4)
	*p-*value	0.633 *	0.004	0.001	0.001
Quality of life	TPG × CG	0.9 (−0.56 to 2.36)	−6.2 (−7.79 to −4.60)	−11.6 (−13.4 to −9.7)	−18.9 (−21.17 to −16.62)
	*p-*value	0.224 *	0.002	0.001	0.001

* Non-significant, TPG—tele-physical therapy group, CG—control group, HbA1c—hemoglobin A1c, FEV1—forced expiratory volume 1, FVC—forced vital capacity, MVV—maximal voluntary ventilation, PEF—peak expiratory flow.

## Data Availability

Data are not publicly available but can be obtained from the corresponding author on request.

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
