# Peer review of "Role of Tele-Physical Therapy Training on Glycemic Control, Pulmonary Function, Physical Fitness, and Health-Related Quality of Life in Patients with Type 2 Diabetes Mellitus (T2DM) Following COVID-19 Infection—A Randomized Controlled Trial"

_healthcare, 2023, doi:10.3390/healthcare11121791_

Round 1

Reviewer 1 Report

Dear authors.

These are my comments:

Brief summary

This manuscript was designed with the aim to determine the clinical effects of tele-physical therapy (TPT) on T2DM following COVID-19 infection. The outcome measures were 31 HbA1c levels, pulmonary function (forced expiratory volume in the first second (FEV1), forced vital 32 capacity (FVC), FEV1/FVC, maximum voluntary ventilation (MVV), and peak exploratory flow 33 (PEF)), physical fitness, and quality of life (QOL).  

Abstract

Abstract is sufficiently written

Introduction

·       It would be good if the author can emphasize the uniqueness of the implementation of tele-rehabilitation on diabetic patients as in this manuscript, as compared the published papers on similar subjects, that can address the current research gaps.

Method

·       Randomization and allocation. A detailed description on the randomization is needed. Any tool used for the randomization, eg random number generator or digital application? Any subject numbering involved?

·       Intervention: Three therapists involved in the study. Any measure implemented to reduce variability between the therapist in term of how they assist the program (provide education etc)? Please explain in detail.

·       Outcome measure: Why the timeframe of monitoring was decided at 8 weeks, 6 months and 12 months? Please add the explanation.

·       Secondary outcome: Short Form 213 Health Survey-12 (SF-12). Why the tool was chosen to measure the QoL in this study? Please describe in detail.

·       Sample size: the formula used was meant for which type of study design, statistical test or objective? Please add the explanation with the reference.

·       Please explain the inclusion and exclusion criteria in the main text under Methodology

 Results

·       “Out of two hundred and forty-seven participants, thirty-six subjects had experience 233 with previous physical therapy interventions”—what does this mean? What are the implications on the results? Please add the explanation.

Discussion

·       In general, the discussion is too superficial. More explanation needed, especially on the potential reason or mechanism of the effect of significant factors on the clinical outcomes.

·       Paragraph 1: most common 305 reason for non-attendance was personal inconvenience. Please explain the examples of personal experience.

·       Paragraph 2: we are the first to study the effects of TPT on patients 314 with T2DM following COVID-19 infection, Why this is important? Please explain.

·       Paragraph 3: no literature available on the effects of tele-328 physical therapy on the pulmonary function. Why this is important? Please explain.

·       Paragraph 4: Nay reason for the contradictory results? Ant potential mechanism related to the results? Please add the explanation

·       Paragraph 5: Lack of reference in this paragraph. Even though tele-physical therapy is proven to be effective as well as cost-effective. Please add reference to support this statement.

·       Paragraph 6: Our findings are generalizable. Generalizable to?

·       Paragraph 6: Third, the treatment provided to the tele-365 physical therapy and control intervention groups were not similar, which could have affected the results. What do the authors meant by this? Ofc course the treatment should not be similar in both group. How it affect the results? Please explain this in detail.

Conclusion 

·       Adequately described

Author Response

Dear Reviewer,

Thank you for your recent review regarding our paper on ‘Role of Tele-Physical Therapy Training on Glycemic Control, Pulmonary Function, Physical Fitness, and Health-Related Quality of Life in Patients with Type 2 Diabetes Mellitus (T2DM) Following COVID-19 Infection—A Randomized Controlled Trial.”

We would like to thank you for the opportunity to submit a revised manuscript. Here is a corrected manuscript with a point-by-point response to comments from your side. We appreciate all the comments, suggestions, and positive criticism raised by you, which were of great value, and certainly helped us to improve the quality of our manuscript. We made all possible efforts to properly address and/or reply to the comments raised. Additionally, a complete review of the article was done for grammar and readability. A significant number of modifications were made to the article after several reviews to not only address your comments, but to correct for any grammatical errors, or readability problems. The authors of this paper greatly thank you for such a great review and fantastic input. The authors greatly appreciate the work of all the reviewers as well as the editors on this manuscript. Many points were a great addition to the quality of this article. Each comment made by a you was addressed point by point below.

Regards

Dr. Gopal Nambi

Reviewer 2 Report

The manuscript entitled Role of Tele-Physical Therapy Training on Glycemic Control, Pulmonary Function, Physical Fitness, and Health-Related Quality of Life in Patients with Type 2 Diabetes Mellitus (T2DM) Following COVID-19 Infection—A Randomized Controlled Trial investigates the impact of tele-rehabilitation in diabetes-related parameters in COVID-19 survivors. It provided important information for post-COVID-19 era, however, additional explanations are needed before publication.

1.     It is not clear what was the inclusion criteria  regarding COVID-19: what was the time frame between COVID-19 infection and recruitment? Were the participants hospitalized or not for COVID-19 infection? Were they having post-COVID symptoms?

2.     Have you observed any difference in tele-rehabilitation effects in type 2  diabetic patients taking metformin in comparison to those who did not since interaction of metformin and exercise indicated greater gains in 6MWT in previous studies?

3.     Have you validated SF-12v2 as instrument in your population sample? What language was used?

4.     Did you have information on previous physical activity of participants? It is known that regular exercise would have more significant impact on HbA1c.

Author Response

(The authors gave the same response as above.)

Reviewer 3 Report

This Paper reports the findings from a prospective, randomised controlled trial investigating the clinical effects of tele-physical therapy on blood glucose control in those living with T2D following Covid 19 infection.

It is unclear of how long the clinical intervention and control went for, also it was unclear if the process of blinding was actually representative of true blinding that was undertaken.  

The study design seems ok but can do with some strengthening by providing more information on the blinding process. Additionally, the duration of the intervention, and more details on the control group i.e. is this standard care? would improve the paper.

Overall, the paper has merit and would be valuable, but requires significant review. Paper is attached with reviewer comments. In additional to the comments above, the following are for suggested improvements:

1. In text referencing needing review.

2. Clarity around the levels of evidence used i.e. reports vs RCTs

3. limitations and strengths of the trial need to be highlighted with greater clairty. 

4. consistency of abbreviation use is needed.

I feel that the clarity of papers English inhibits my assessment and why i feel it is important to have the paper revised and reassessed.

The attached document has quite a few comments on the use of English and some suggested corrections for you and the team to consider. 

Please refer to attached document for feedback. Thank you.

Author Response

(The authors gave the same response as above.)

Round 2

Reviewer 1 Report

Dear authors,

I have no further comments

Reviewer 3 Report

Thank you for your review and positive turn around on the manuscript. The adjustments that you and the team have made have certainly improved the readability of the manuscript in a way that represents what was good science. Thank you and all the best.